# Immunity Induced by Inactivated SARS-CoV-2 Vaccine: Breadth, Durability, Potency, and Specificity in a Healthcare Worker Cohort

**DOI:** 10.3390/pathogens12101254

**Published:** 2023-10-18

**Authors:** Ying Chen, Caiqin Hu, Zheng Wang, Junwei Su, Shuo Wang, Bin Li, Xiang Liu, Zhenzhen Yuan, Dan Li, Hong Wang, Biao Zhu, Yiming Shao

**Affiliations:** 1Department of Infectious Diseases, Zhejiang Hospital, Hangzhou 310003, China; 21718041@zju.edu.cn; 2State Key Laboratory for Diagnosis and Treatment of Infectious Diseases, National Clinical Research Center for Infectious Diseases, National Medical Center for Infectious Diseases, Collaborative Innovation Center for Diagnosis and Treatment of Infectious Diseases, The First Affiliated Hospital, Zhejiang University School of Medicine, Hangzhou 310003, China; hucaiqin@zju.edu.cn (C.H.); zjusujunwei@zju.edu.cn (J.S.); liuxiang0202@zju.edu.cn (X.L.); 3National Key Laboratory of Intelligent Tracking and Forecasting for Infectious Diseases, National Center for AIDS/STD Control and Prevention, Chinese Center for Disease Control and Prevention, Beijing 102206, China; wangzhengcdc@163.com (Z.W.); wswangshuo1029@126.com (S.W.); libincdc@163.com (B.L.); zzzm18981996@163.com (Z.Y.); 4Changping Laboratory, Beijing 102206, China

**Keywords:** SARS-CoV-2, vaccination, antibody response, cellular immunity, healthcare workers

## Abstract

Vaccination has proven to be highly effective against severe acute respiratory syndrome coronavirus 2 (SARS-CoV-2), but the long-term immunogenicity and the functional preserved immune responses of vaccines are needed to inform evolving evidence-based guidelines for boosting schedules. We enrolled 205 healthcare workers into a cohort study; all had received three doses of BBIBP-CorV (China Sinopharm Bio-Beijing Company, Beijing, China) inactivated vaccine. We assessed SARS-CoV-2 specific binding antibodies, neutralizing antibodies, and peripheral T and B cell responses. We demonstrated that more robust antibody responses to SARS-CoV-2 were elicited by booster immunization compared with primary vaccination. Neutralizing antibody titers to SARS-CoV-2 Omicron BA.1 were also efficiently elevated post-homologous vaccine booster despite being in a lower titer compared with the prototype stain. In addition to S-specific humoral and cellular immunity, BBIBP-CorV also induced N-specific antibody and effector T cell responses. The third-dose vaccination led to further expansion of critical polyfunctional T cell responses, likely an essential element for vaccine protection. In particular, a functional role for Tfh cell subsets in immunity was suggested by the correlation between both CD4^+^ Tfh and CD8^+^ Tfh with total antibody, IgG, B cell responses, and neutralizing antibodies. Our study details the humoral and cellular responses generated by the BBIBP-CorV booster vaccination in a seven-month follow-up study. There is a clear immunologic boosting value of homologous inactivated SARS-CoV-2 vaccine boosters, a consideration for future vaccine strategies.

## 1. Introduction

The global pandemic caused by SARS-CoV-2 has persisted since its recognition in December 2019. Due to continuously emerging immune escape variants, the necessity of booster vaccinations for coronavirus disease 2019 (COVID-19) is apparent. Various technologies have been used for vaccine delivery, such as mRNA, DNA, inactivated, recombinant protein, and adenovirus-based vectors. The BBIBP-CorV (China Sinopharm Bio-Beijing Company, Beijing, China) is an inactivated vaccine approved for registration and emergency use. A theoretical advantage of BBIBP-CorV is that, unlike other popular vaccines carrying the spike (S) epitopes only, inactivated vaccines retain the integrity of the virus particle envelopes, providing immune exposure to a wider range of epitopes. N protein, for example, shows cross-reactivity between coronaviruses and also can induce N-antibody responses in COVID-19 patients [1,2].

Neutralizing antibody responses against SARS-CoV-2 correlate with protection efficiency [3,4,5]. The waning of immunity after vaccination corresponds to the increasing risk of breakthrough infections of SARS-CoV-2. Emerging variants of concern (VOC) have included Alpha (B.1.1.7), Beta (B.1.351), Gamma (P.1), Delta (B.1.617.2), and Omicron (B.1.1.529) variants. Omicron has further rapidly evolved many subvariants (BA.1, BA.1.1, BA.2, BA.2.12.1, BA.4, BA.5, BF7, BQ, XBB, EG.5, etc.) over time and is the dominant circulating strain globally [6,7,8]. The higher immune escape capacity and greater transmissibility of the Omicron variants has greatly increased the number of breakthrough infections [9,10,11,12]. While the booster dose of the vaccine can improve immunogenicity of the vaccine series, vaccine effectiveness against the Omicron variant remains unclear.

In addition to the humoral immune response, the vaccine-induced cellular immunity is helpful in controlling viral infection. B cells promote the T cell differentiation into T follicular helper (Tfh) cells, improving humoral immune responses [13,14]. Specific T cell responses help control SARS-CoV-2 replication, reducing COVID-19 disease severity [15,16,17].

After inactivated vaccination, the nature and the differentiation state of antigen-specific memory and effective T and B cells remain to be elucidated. For example, it is still unclear whether the CD4^+^ and CD8^+^ Tfh cells can be boosted and whether these cells correlate with memory B cells and neutralizing antibodies. It remains unknown, too, as to how long the subsets of memory cells last and how these cells contribute to long-term immunological memory and protective immunity. In this study, we sought to define the differentiation state of immune cells and address these questions following inactivated vaccine prime and boost in healthcare workers who had received inactivated vaccines.

## 2. Results

### 2.1. A Longitudinal Cohort of Vaccinees Immunized by BBIBP-CorV

All 205 participating healthcare workers from Zhejiang hospital had received three doses of BBIBP-CorV, with a three-week interval between the first and second vaccine doses and an average of 274 days between the second and third vaccine doses. This cohort included 66 men and 139 women; 138 workers were under the age of 40 and 67 workers were 40 or older (Table 1).

### 2.2. Robust Antibody Responses to SARS-CoV-2 Elicited by Booster Immunization

We detected the neutralizing antibody, total antibody, IgG and IgM by chemiluminescence immunoassay assays, nucleoprotein (N) antibody, and receptor-binding domain (RBD) antibody of SARS-CoV-2 by ELISA in plasma samples at all follow-up time points. The antibody magnitude and the seropositivity of SARS-CoV-2 after vaccination are shown in Figure 1 and Table 2.

Both the peak and durable antibody responses induced by the inactivated vaccine booster dose were significantly stronger than those induced by the second dose. The descending speed of antibody levels after the third dose were also significantly slower than those of the second dose.

Unlike other vaccines, in addition to the potent spike antibody, the N antibody response can also be induced by the BBIBP-CorV vaccine. We found that the seropositivity of the N antibody response was 31.3% after two doses of the vaccine. After booster vaccination, the seropositivity of the N antibody reached 98.8%, and remained at 92.4% after 7 months.

### 2.3. Higher, More Durable, and Broadly Neutralizing Antibody Titers Induced by Booster Immunization

We found neutralizing antibody titers against the wildtype strain and Omicron BA.1 of SARS-CoV-2 both by pseudovirus and authentic viral neutralization assays in parallel comparison. The GMTs of the neutralization titers to wildtype after the third vaccination were about three times more than those after the second vaccination (Figure 2). No neutralizing antibodies against the Omicron variant were induced after only two doses of vaccine. The seropositivity of neutralizing antibodies against Omicron BA.1 was 74–86% two weeks after the third vaccination and 38–44% seven months after the third vaccination.

Figure 2C shows the strong correlations between neutralization titers by pseudovirus and authentic viral neutralization assays. Consistently, data in Figure 2D demonstrate positive correlations between neutralization titers against wildtype and Omicron BA.1.

### 2.4. Specific Effector T Cells, Follicular Helper T Cells, and B Cell Immunity Induced by SARS-CoV-2 Boosters

The follicular helper T cells (Tfh) are involved in the humoral response by activating the germinal center (GC) B cells to differentiate into antibody-secreting plasma cells and memory B cells. The phenotypic characteristics of Tfh cells were investigated using CXCR5^+^PD-1^+^ markers in this study. The median frequencies of Tfh cells (0.985% and 1.445% of CD4^+^ and CD8^+^ Tfh) peaked two weeks after the booster vaccination, with significant differences seen between groups (Figure 3A). The frequency of CD19^+^CD20^+^ and CD71^+^ were the highest 2 weeks after vaccination, then decreased significantly 4 months and 7 months after vaccination (Figure 3B).

We performed enzyme-linked immunospot assay (ELISPOT) to detect IFN-γ levels in PBMC cells. The average number of S1-specific IFN-γ effector T cells were 24, 103, and 29 per million PBMCs, and the average number of N-specific IFN-γ effector T cells were 17, 86, and 40 per million PBMCs at the time points before booster vaccination, two weeks after, and seven months after booster vaccination, respectively. In addition, both S1 and N-specific effector T cells 2 weeks after vaccination were significantly higher than those before vaccination and 7 months after booster immunization (Figure 3C). Two weeks after vaccination, about one-third of healthcare workers could elicit S1- and N-specific effector T cells; only 7.2% and 24.6% of the population preserved S1- and N-specific effector T cells seven months after vaccination, respectively (Figure 4). Thus, Tfh, B cells, and specific effector T cells peaked at 2 weeks or 4 weeks, and gradually decreased after the booster vaccination.

### 2.5. Synergistic Effect of Antibody and Cellular Immunity after Vaccination

We further analyzed the association between antibodies and cellular response (Figure 5A). Positive correlations were seen between CD19^+^CD20^+^ and CD71^+^, between CD4^+^ Tfh and CD8^+^ Tfh, between the N-specific and S1-specific effector T cells, and between B cells and Tfh. CD19^+^CD20^+^, CD71^+^, and CD4^+^ Tfh cell subsets showed significant positive correlations with the neutralizing antibody to the SARS-CoV-2 wildtype. Also, S1-specific T cells and CD71^+^ showed significant positive correlations with the neutralizing antibodies to the SARS-CoV-2 Omicron BA.1 variant (Figure 5B,C). CD4^+^ Tfh cells showed significant positive correlations with the neutralizing antibody to SARS-CoV-2 (Figure 5D). N antibody levels were also significantly associated with the neutralizing antibody and N-specific effector T cells.

## 3. Discussion

The SARS-CoV-2 vaccine has significantly reduced the severity and mortality of COVID-19 [5,18]. However, due to the fading of the immune protection effect and the emergence of new variants, breakthrough symptomatic infection cases are common. Our longitudinal cohort followed 205 healthcare workers for 16 months, and for 7 months after the third vaccine (booster). Not only humoral antibodies and cellular immunity but also short-term and durable immunity were examined. About 70% of the vaccinees retained neutralizing antibody titers against Omicron BA.1 at 2 weeks and 40% at 7 months post-booster vaccine; no neutralizing antibody capacity against the Omicron variant was detected after only the second dose of vaccine. The booster dose vaccination could induce more potent, durable, and broad antibody responses and neutralization capabilities than those elicited by the second dose of immunization, which was consistent with prior work [12,19].

In order to analyze whether a vaccine booster is required, the immune response to SARS-CoV-2 is essential. CD4^+^ and CD8^+^ T cell responses are generated rapidly, with a delayed humoral immunity after breakthrough infection [20,21]. Understanding antigen-specific T cell response can enable a deeper understanding of the function and heterogeneity of the pathogen-specific response. The common assays used to detect and characterize antigen-specific T cells include the enzyme-linked immunospot assay, intracellular staining (ICS), activation-induced markers (AIMs) assay, and T cell antigen receptor (TCR) repertoire of the single cells. We discovered S1- and N-specific T cell protective responses induced by BBIBP-CorV boosting. TH1 or TH2 cell responses cannot be induced by inactivated vaccines in non-human primates and human individuals after only a primary vaccination [22,23]. In contrast, we found that S1-/N-specific effector T cells of SARS-CoV-2 were elicited after the booster vaccination. It is likely that weak specific T cell responses induced by the primary vaccination were enhanced substantially after the booster vaccine dose. Lower S-specific T cell immune responses were noted from inactivated vaccines than from the mRNA vaccine. However, it was found that the multi-antigen CD4^+^ T cell response induced by the inactivated vaccine may be more protective in improving disease severity [24]. Multi-antigen specific T cell immune responses have also been found in mild and asymptomatic SARS-CoV-2-infected patients, and an antigen-specific response represents the immune protection from structural protein-specific T cells [25,26]. Compared with S-specific T cells, N-specific T cells induced by inactivated vaccinees are more durable to tolerate mutations characterizing the Omicron viral lineage [24].

We found very consistent dynamics and correlations between cellular and humoral immune responses after the booster dose. These results suggest that cellular immunity can provide protective effect indirectly, consistent with previous studies [27,28]. The protective T cell response (CD4^+^ and CD8^+^ T cells producing interferon IFN-γ, commonly referred to as a “type 1” immune response) against SARS-CoV-2 infection has been noted by others [29,30,31]. Ineffective IFN-γ innate immunity has been associated with a failure to control a primary SARS-CoV-2 infection and a high risk of fatal COVID-19 [32,33,34]. Our findings suggest that antibody and cellular immunity may demonstrate synergistic effects in the control of SARS-CoV-2 infection.

This cohort study was a major strength of our study that deployed a wide swath of immunological response indicators. Several limitations were extant. First, while we analyzed the specific T cells of SARS-CoV-2, specific B cell immunity against the wildtype and variants was not studied. Second, PBMCs were not collected and isolated at the earlier follow-up visits, and cellular immunity was not seen after the first and second doses of vaccine.

The second booster vaccine was implemented rapidly and widely in some countries and was deemed inadequate to prevent symptomatic Omicron infection [35,36,37]. Immune responses induced by the secondary booster vaccine were reported to be no higher than those induced by the first booster dose. Immune attenuation of the booster vaccine was significant slower than that seen in the two-dose vaccine in this study [38]. The frequency of the second booster vaccine should be based on the substantial and sustained data rather than on short-term immunity. Also, the cross-immunity vaccine supply worldwide should be taken into consideration.

We found that an inactivated vaccine booster dose (third dose) induced substantial potent, durable, and broad immunity, including N-specific immunity. Our data are compatible with the conclusion that humoral and cellular immunity may be synergistic in the control of SARS-CoV-2 infection. To reduce severity of infection, hospitalization, and death from SARS-CoV-2 infection, the booster vaccination should be rigorously promoted and implemented.

## 4. Materials and Methods

### 4.1. Study Participants

All the participants were healthcare workers from Zhejiang Hospital who had never been infected with SARSR-CoV-2 during the entire follow-up during the period from December 2020 to April 2022. All participants had been immunized with 3 doses of vaccines; the interval was 3 weeks between the 1st and 2nd doses for nearly all participants. The 3rd dose of vaccination was received an average of 274 days (range 146–291 days) after the 2-dose vaccination. We drew blood from participants at the following times: 2 weeks after 1st BBIBP-CorV immunization; 2 weeks and 6 months after the 2nd immunization; and 2 and 4 weeks and 4 and 7 months after the 3rd immunization (homologous booster). The study was conducted in accordance with the Declaration of Helsinki, and the protocol was approved by the Ethics Committee of Zhejiang Hospital (Reference Number 2021-30K-X1). Written informed consent was obtained from all participants.

### 4.2. Enzyme-Linked Immunosorbent Assay (ELISA)

We described methodological details previously [39]. We measured anti-RBD and anti-N antibodies of SARS-CoV-2 using SARS-CoV-2 RBD/N ELISA kits in accordance with manufacturers’ instructions (Wantai Biological Pharmacy, Beijing, China). The detailed information has been provided in a previous article [39].The reaction was stopped by adding 50 µL of 1 nm H_2_SO_4_ to each well, and the reading was taken at 450 nm.

### 4.3. Chemiluminescent Microparticle Immunoassay (CMIA)

The SARS-CoV-2 Abs (neutralizing antibodies, total antibody, anti-IgG, anti-RBD, and anti-N) detection kits (Maccura Biosystem Co., Sichuan, China) were based on chemiluminescent microparticle immunoassay (CMIA). We mixed a 10 µL plasma sample, 50 µL magnetic beads, and a 50 µL buffer, incubating for 10 min in a reaction cup. We then added 100 µL acridine ester-labeled marker to resuspended magnetic beads that were incubated for 10 min and then washed. We determined the luminescence signal value after adding the substrate solution using a matched automatic chemiluminescence immunoassay analyzer. We calculated the concentrations of neutralizing antibodies according to the standard calibration curve; values > 6 AU/mL were considered positive. We presented the results of total antibodies, IgG, and IgM as the S/CO (absorbance of sample/cutoff of calibration); S/CO < 1 was considered positive and S/CO ≥ 1 negative.

### 4.4. Pseudovirus Neutralization Test (PVNT)

We generated the pseudovirus by co-transfection of HEK 293 T cells with pcDNA3.1-S-COVID19 and pNL4-3Luc, which carry the optimized spike (S) gene and a human immunodeficiency virus type 1 (HIV) backbone, respectively. We added 150 µL serial dilutions of human sera (4 serial 3-fold dilutions in Dulbecco’s minimum essential medium (DMEM) with an initial dilution 1:20) into 96-well plates. We then added 50 µL pseudovirus of SARS-CoV-2 with a concentration of 1300 TCID50/mL to the plates, incubating them at 37 °C for 1 h. We added Hu-h7 cells to the plates (1.5 × 10^4^ cells/100 uL cells per well), incubating them at 37 °C in a humidified atmosphere with 5% CO_2_. We performed chemiluminescence detection after 48 h incubation. The Reed–Muench method was used to calculate the virus neutralization titers [40]. The result reported as a half-maximal inhibitory concentration of PVNT (PVNT50).

### 4.5. Authentic Viral Neutralization Test (AVNT)

The authentic viral neutralization test of SARS-CoV-2 was performed as detailed previously [41]. In brief, we measured the neutralizing antibody titers against the wildtype strain and the variants (Beta B.1.1.7, Gamma P.1, and Delta B.1.617) in serum by using a cytopathic effect-based microneutralization assay in Vero cells (National Collection of Authenticated Cell Cultures, National Academy of Science, Beijing, China). We then mixed serum with the same volume of viral solution to achieve a final concentration of 100 TCID50 per well. The reported titer was the reciprocal of the highest sample dilution that protected at least 50% of cells from cytopathic effects. Serum dilution for the neutralization assay started from 1:4, and seropositivity was defined as titer ≥ 1:4.

### 4.6. Flow Cytometry

To measure antigen-specific effector T cells, we performed intracellular staining (ICS) using 15-mers S1 and N peptide pools (overlapping by 11 amino acids) from SARS-CoV-2 strain (Bio-scientific Co., Shanghai, China). We stimulated peripheral blood mononuclear cells (PBMCs) isolated from the subjects with S1 and N peptide pools (2 μg/mL) in the presence of Brefeldin A (Sigma) (100 ug/mL) for 6 h. We used dimethyl sulfoxide (DMSO, Sigma) as a negative control for peptides. As a positive control, cells were stimulated with staphylococcal enterotoxin B (SEB) from staphylococcus at a concentration of 2 ug/mL. After stimulation, we washed the cells with phosphate-buffered saline (PBS) and stained them with an ultraviolet excitable amine-reactive viability dye (LIVE/DEAD, Invitrogen) to exclude dead cells. After 20 min of incubation, we further washed the PBMCs with PBS and stained them with anti-CD4-PE-CF594 (RPA-T4), anti-CD8-Pacific Blue (RPA-T8), anti-CCR5-BB700 (RF8B2), anti-CD56-BV650 (HCD56), anti-CCR7 (G043H7), anti-CD45RA-APC (HI100), anti-CD19-BV510 (SJ25C1), and anti-PD-1-BV605 (EH12.2H7). After an additional 20 min of incubation, we again washed the PBMCs with PBS, fixed them with 1*BD FACS lysing (BD) for 10 min, and ruptured their cytomembranes with 0.25% saponin. We then stained the PBMCs with anti-CD3-BV570 (UCHT1), anti-CD154-PE-Cy7 (24–31), anti-IFN-γ-A700 (B27), anti-TNF-α-FITC (MAb11), and anti-IL-2-PE (MQ1-17H12) for 30 min. The spike-/N-stimulated group subtracted-negative control data (>0.05% for CD4^+^ T cells and CD8^+^ T cells) were defined as the specific T cells.

To measure the B cells, follicular helper T cells, and effector memory T cells, we stained PBMCs with anti-CD4-BV786 (SK3), anti-CD8-APC (RPA-T8), anti-CD14-BV421 (MφP9), anti-CD20-PE-CF594 (2H7), anti-IgG-BV605 (G18-145), anti-CD95-FITC (DX2), anti-CD71-PE (M-A712), anti-IgM-PercpCy5.5 (G20-127), anti-CD27-APC-Cy7 (M-T271), anti-CXCR5-APC-R700 (RF8B2), anti-CXCR3-PECy5 (1C6), anti-CD45RA-BV650 (HI100), anti-CD19-BV510 (SJ25C1), anti-CD3-BV570 (UCHT1), and anti-PD-1-PECy7 (eBioJ105). We obtained flow cytometry data using a Fortessa LSR flow cytometer (LSRFortessaTM, BD) and performed data analysis using FlowJoTM (TreeStar, Ashland, OR, USA).

### 4.7. ELISPOT Assay

We ran IFN-γ ELISpot assays according to the manufacturer’s instructions (BD, Cat. 349202). Briefly, 100 µL of diluted anti-IFN-γ solution was added to each well and stored at 4 °C overnight. Plates were washed with blocking solution (RPMI 1640 with 10% fetal bovine serum, 1% penicillin/streptomycin/L-glutamine) and blocked for 2 h at room temperature. After incubation and plate washings, 0.2 million PBMCs per test with SARS-CoV-2 S1 and N peptide pools at a final concentration of 2 ug/mL were added per well and incubated for 18 h. Then, diluted biotinylated IFN-γ antibody was added per well and incubated for 2 h at room temperature. After that, the diluted enzyme conjugate (streptavidin-HRP) was added to each well and incubated for 1 h. After, 100 µL of substrate solution was added per well for about 15 min and washed with deionized water. The spots on the plates were analyzed by an ELISPOT plate reader (AID Gmbh, Strassberg, Germany). To quantify antigen-specific responses, we subtracted mean spots of the DMSO control wells from the peptide-stimulated wells and expressed the results as spot-forming units (SFU) per 10^6^ PBMCs. We considered results > 20 SFU/10^6^ PBMCs following control subtraction as positive.

### 4.8. Statistical Analysis

Data and statistical analyses were performed in Prism (version 8.0.2), Origin2021b (version 9.8.5) and SPSS software (version 23.0), unless otherwise stated. Two-tailed nonparametric Mann–Whitney U test and Kruskal–Wallis test were performed on numerical data. *p*-values < 0.05 were considered statistically significant.

## Figures and Tables

**Figure 1 pathogens-12-01254-f001:**
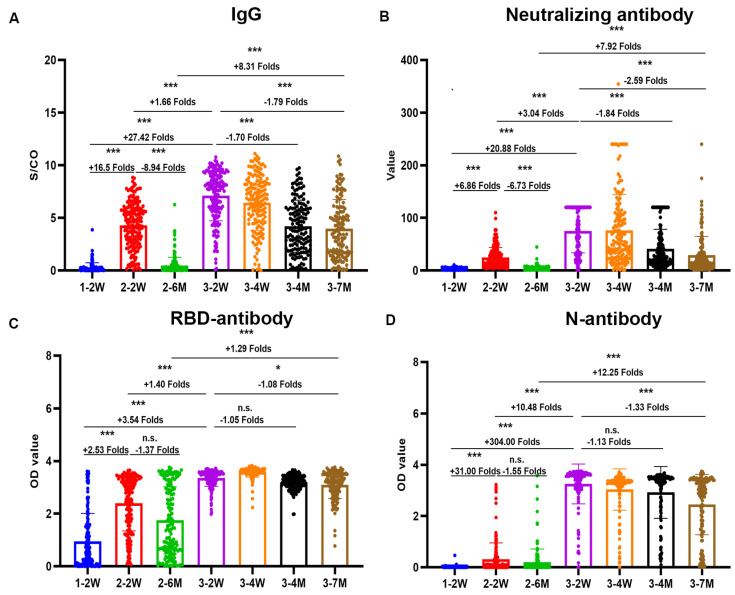
Dynamic antibodies antibody responses induced by BBIBP-CorV. IgG (**A**), neutralizing antibody (**B**) by chemiluminescence immunoassay assays, receptor-binding domain (RBD) antibody (**C**), and nucleoprotein (N) antibody (**D**) of SARS-CoV-2 by enzyme-linked immunosorbent assay at seven cohort time points. Scatter points on the bar represent a single record. The bars and lines indicate mean and standard deviation, respectively. Fold changes are noted on top of the bar. Two-tailed nonparametric Dunn’s Kruskal–Wallis test was used for multiple comparisons. * *p* < 0.05, *** *p* < 0.001. n.s. no significance.

**Figure 2 pathogens-12-01254-f002:**
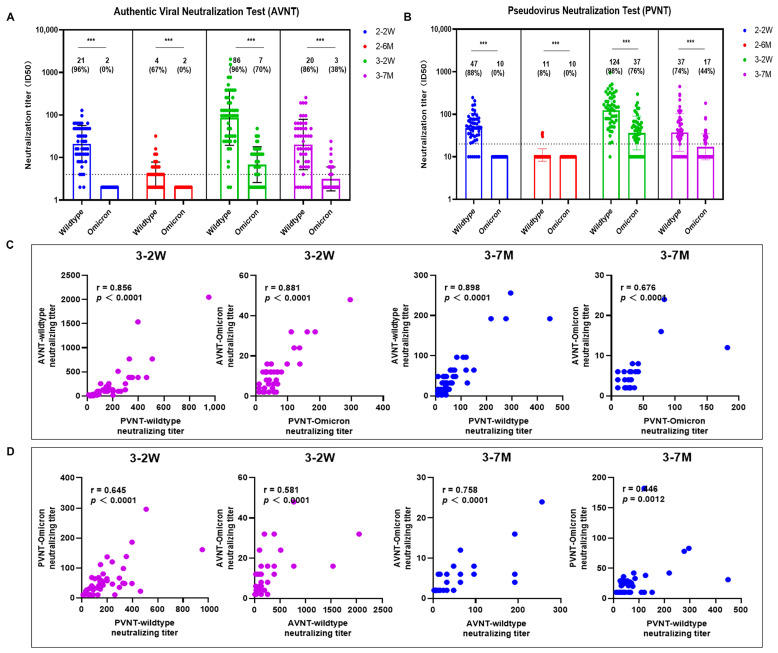
Neutralizing antibody titers against SARS-CoV-2 wildtype and Omicron BA.1. The neutralizing antibody titers of SARS-CoV-2 were detected against SARS-CoV-2 wildtype and Omicron BA.1 both by authentic viral neutralization (AVNT) and pseudovirus neutralization tests (PVNT) at four different time points (**A**,**B**). (**C**) shows the correlations between neutralization titers by pseudovirus and authentic viral neutralization assays. (**D**) shows correlations between neutralizing antibody titers against wildtype and Omicron BA.1. Scatter points on the bar represent a record. The bars and lines indicate geometric mean titers (GMT) and standard deviation, respectively. GMT and seropositivity are presented on top of the bar. Two-tailed nonparametric Mann–Whitney U test was used for comparisons between the neutralization titers against wildtype and Omicron variants. *** *p* < 0.001.

**Figure 3 pathogens-12-01254-f003:**
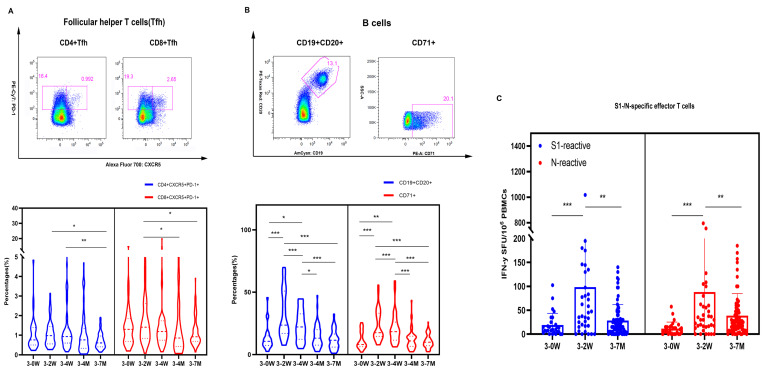
Specific effector T cells, follicular helper T cells, and B cells immunity induced by SARS-CoV-2 booster vaccination. (**A**) presents the gating strategy of CD4^+^/CD8^+^ Tfh expressing CXCR5^+^PD-1^+^ and the frequency of CD4^+^ and CD8^+^ Tfh cells. (**B**) presents our gating strategy of mature B cells expressing CD19^+^CD20^+^ and active B cells expressing CD71^+^. The S1-/N-specific IFN-γ effector T cells level of SARS-CoV-2 were detected by ELISPOT (**C**). We used 2-tailed, nonparametric Dunn’s Kruskal–Wallis tests for multiple comparisons. * *p* < 0.05, ** *p* < 0.01, *** *p* < 0.001.

**Figure 4 pathogens-12-01254-f004:**
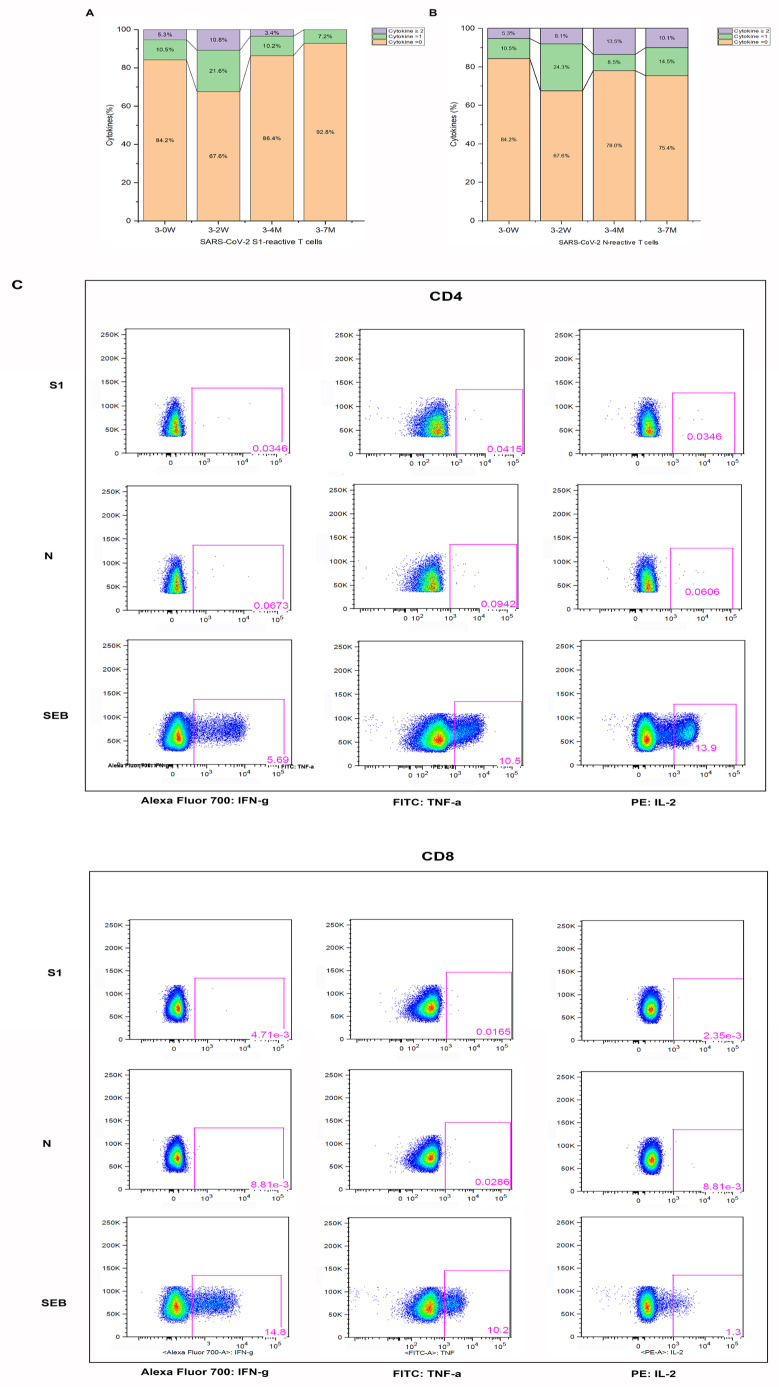
Effector T cells specific to SARS-CoV-2 by intracellular staining (ICS) for cytokines. The percentage of effector T cells to specific S1 (**A**) and N (**B**) are presented for the different time points. The representative gating strategy is presented for CD4^+^/CD8^+^ effector T cells expressing cytokines (IFN-y, TNF-a, IL-2) detected in vaccinees (**C**) stimulated with SARS-CoV-2 specific S1; N peptide pools and staphylococcal enterotoxin B (SEB) from staphylococcus are presented in (**C**).

**Figure 5 pathogens-12-01254-f005:**
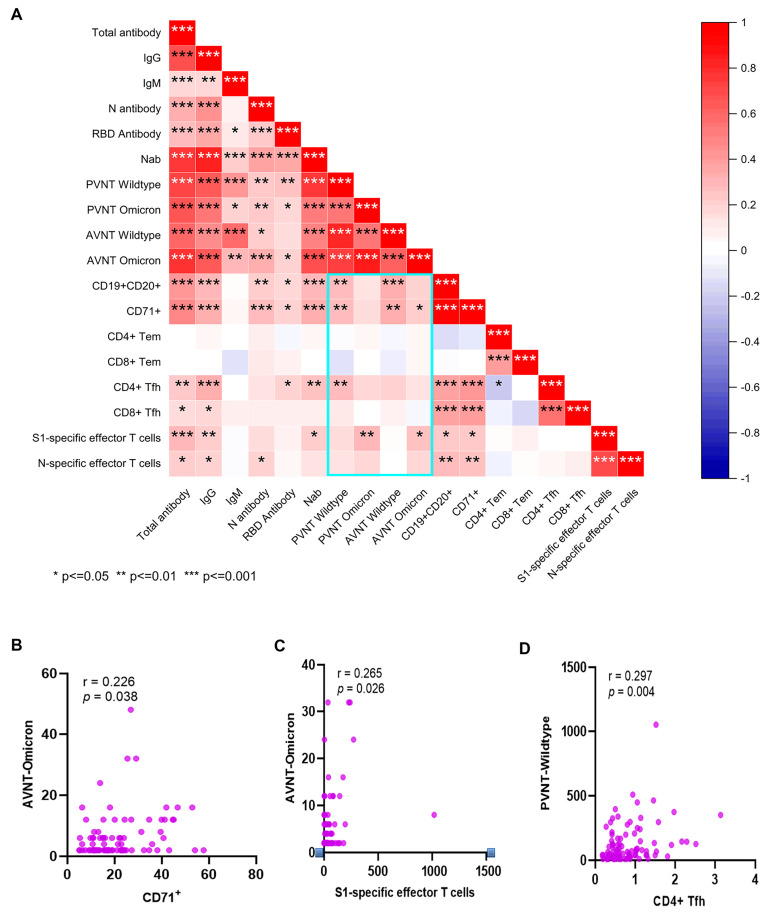
The correlation between humoral and cellular immunity. Heat map of Pearson correlation matrices are shown between cellular and humoral immunity after booster immunization. Statistically significant correlations are indicated with an asterisk (*) (**A**). Correlations are shown between the CD71^+^ or S1-specific effector T cells and neutralizing antibody to the SARS-CoV-2 Omicron BA.1 variant (**B**,**C**), and between the CD4^+^ Tfh cells and neutralizing antibody to SARS-CoV-2 (**D**). A scatter point represents a record. R represents the degree of linear deviation obtained by fitting the experimental data, and *p*-value tests assess statistical significance of regression equations. * *p* < 0.05, ** *p* < 0.01, *** *p* < 0.001.

**Table 1 pathogens-12-01254-t001:** Characteristics and the sampling time points after immunization.

Characteristic	Participants (N = 205; 100%)
Sex	
Male	66 (32.2%)
Female	139 (67.8%)
Age (years)	
<40	138 (67.3%)
≥40	67 (32.7%)
Follow-up time after immunization *	
1-2W	147
2-2W	194
2-6M	190
3-2W	163
3-4W	154
3-4M	148
3-7M	144

* 1-2W—two weeks after first immunization; 2-2W—two weeks after the second immunization; 2-6M—six months after the second immunization; 3-2W—two weeks after the third immunization; 3-4W—four weeks after the third immunization; 3-4M—four months after the third immunization; 3-7M—seven months after the third immunization.

**Table 2 pathogens-12-01254-t002:** The antibody magnitude and seropositivity of SARS-CoV-2 after vaccination.

Sampling Time *	Neutralizing Antibody Titers		Total Antibody Titers		IgG Titers		Nucleoprotein (N) Antibody Titers		RBD Antibody Titers		IgM Titers	
	Mean	Seropositivity %	Mean	Seropositivity %	Mean	Seropositivity %	Mean	Seropositivity %	Mean	Seropositivity %	Mean	Seropositivity %
1-2W	3.61	8.28	1.92	37.2	0.26	8.28	0.01	0.68	0.95	68.7	0.36	6.21
2-2W	24.8	90.7	17.4	90.7	4.29	90.7	0.31	31.3	2.40	97.8	3.16	52.1
2-6M	3.68	20.0	4.24	72.1	0.48	12.6	0.20	20.8	1.75	88.1	0.09	1.05
3-2W	75.4	96.9	236.5	100	7.13	98.2	3.25	98.8	3.36	100	0.30	6.13
3-4W	76.5	94.8	90.2	100	6.42	96.7	3.04	99.3	3.58	100	0.31	5.23
3-4M	41.0	96.0	35.1	100	4.20	91.9	2.92	98.7	3.19	100	0.12	2.70
3-7M	29.2	86.8	23.5	96.5	3.99	86.8	2.45	92.4	3.10	100	0.10	1.39

* 1-2W—two weeks after first immunization; 2-2W—two weeks after the second immunization; 2-6M—six months after the second immunization; 3-2W—two weeks after the third immunization; 3-4W—four weeks after the third immunization; 3-4M—four months after the third immunization; 3-7M—seven months after the third immunization.

## Data Availability

The datasets and materials generated during and/or analyzed during the current study are available from the corresponding authors on reasonable request.

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
