# Peer review of "Immunity Induced by Inactivated SARS-CoV-2 Vaccine: Breadth, Durability, Potency, and Specificity in a Healthcare Worker Cohort"

_pathogens, 2023, doi:10.3390/pathogens12101254_

Round 1

Reviewer 1 Report

At line 69. By this time it seems not accurate to say '...remain unknown'. There are quite a lot reports on protection correlates.

Figure 2B lacks the baseline distinguishing the true and false positive. 

At line 138. 'by triggering GC B cells' sounds not pretty accurate. I would suggest using 'driving' or 'activating'.

Suggest enlarging the font in figures 1 - 4. 

I can't understand the meaning of the gradient color in Figure 5.

At line 220. This inference is of much less significance. There are many scientific reports on antibody and cellular immunity against SARS-CoV-2. 

I cannot understand the sentences at lines 223 - 226. 

At line 243. Please cite references that verified the protectiveness of BBIBP vaccine-induced N-specific T cell response for better understanding the synergistic protection. Authors just presumed a kind of synergistic vaccine protection, hence not a rigorous and clear scientific conclusion yet. The use of 'may be' should not be present in a conclusion. 

can be improved. 

Author Response

Responses to Review

Point: At line 69. By this time it seems not accurate to say '...remain unknown'. There are quite a lot reports on protection correlates.

Response: Thank you for your useful advice. We agree with your comment to our manuscript. Yes,the reports on protection correlates had been extensively explored. During the revision of the draft, we identified this sentence as redundant in the article and consequently excluded it.

Point: Figure 2B lacks the baseline distinguishing the true and false positive. 

Response: The baseline of Figure 2B was added in the manuscript.

Point: At line 138. 'by triggering GC B cells' sounds not pretty accurate. I would suggest using 'driving' or 'activating'.

Response: We had changed the 'by triggering GC B cells' to the 'by activating GC B cells'

Point: Suggest enlarging the font in figures 1 - 4. 

Response: The figures in the article had been renewed.

Point: I can't understand the meaning of the gradient color in Figure 5.

Response: The figure 5 showed a heat map of Pearson correlation matrices. Statistically significant correlations are indicated with asterisk (*). The color gradient represents the R-value of the correlation coefficient in the heat map, and the red color represents the positive correlation and the blue color represents the negative correlation.

Point: At line 220. This inference is of much less significance. There are many scientific reports on antibody and cellular immunity against SARS-CoV-2. 

I cannot understand the sentences at lines 223 - 226. 

Response: We are sorry that we had put paragraphs in the sentences of lines 220-226 by mistake. So, in the revised manuscript, we put these sentences at the second paragraph of discussion part to make it more accurate and easier for you to understand.

Point: At line 243. Please cite references that verified the protectiveness of BBIBP vaccine-induced N-specific T cell response for better understanding the synergistic protection. Authors just presumed a kind of synergistic vaccine protection, hence not a rigorous and clear scientific conclusion yet. The use of 'may be' should not be present in a conclusion. 

Response: Yes, your suggestions are pretty helpful. The previous studies had verified that humoral and cellular immunity can provide protective effect against infection. Also, multi-antigen T cell (including N-specific T cell) response induced by inactivated vaccine presented more protective in improving disease severity. For lacking protective correlation experiments in this study, and which also a limitation existed in this article. So, we just presumed a synergistic effect between humoral and cellular immunity.

Reviewer 2 Report

In this work Ying Chen et al clearly demonstrated the ability of homologous vaccination with inactivated SARS-CoV-2 vaccine BBIBP-CorV to stimulate humoral and cellular immune response against wild type and omicron SARS-CoV-2 variant.

The interesting study present results from different arms of immune response, corroborated with appropriate controls, analysis at different time points after one, two and three immunizations for humoral immunity, and after the third dose for the cellular response. Honestly, authors cited limitations of their study.

Some suggestions can further improve the quality of the work:

-        Update the bibliography with recent publications (no articles of 2023 were cited).

-        The licensed mRNA vaccine against variant can be cited in the introduction.

-        Update the list of circulating variants, supported by bibliography.

-        Table 1 can be implemented with abbreviations of different groups as reported in table 2.

-        Increase the size of all figures. Some written are not readable.

-        Clarify the Y axis in Fig. 1A and B.

-        Detail as the antibody titer was calculated. This data is missing in the cited ref indicated in the text.

-        In fig. legend 1 specify if “antibody” refers to IgG.

-        Specify which variant of omicron was tested.

-        In fig. 2C-D some values are not readable for the presence of some dot.

-        Specify how flow cytometry gates were defined.

-        Specify concentration of BFA and SEB in materials and methods.

-        At line 265 use “SARS-CoV-2 Abs” instead of COVID-19 (that refers to the disease).

-        At line 329 I suppose that “RPMI 1640” was the correct media. If yes, please correct.

-        Detail how spot of ELISPOT plated were read.

Author Response

Comments and Suggestions for Authors

In this work Ying Chen et al clearly demonstrated the ability of homologous vaccination with inactivated SARS-CoV-2 vaccine BBIBP-CorV to stimulate humoral and cellular immune response against wild type and omicron SARS-CoV-2 variant.

The interesting study present results from different arms of immune response, corroborated with appropriate controls, analysis at different time points after one, two and three immunizations for humoral immunity, and after the third dose for the cellular response. Honestly, authors cited limitations of their study.

Some suggestions can further improve the quality of the work:

-        Update the bibliography with recent publications (no articles of 2023 were cited).

Response:  Thank you for your comments on the manuscript. We had updated the references published in 2023.

-        Update the list of circulating variants, supported by bibliography.

Response: The list of circulating variants had updated in the article.

-        Table 1 can be implemented with abbreviations of different groups as reported in table 2.

Response: The abbreviations of different groups had implemented in table 1.

-        Increase the size of all figures. Some written are not readable.

Response: Thank you for your useful advice. The figures in the article had been renewed.

-        Clarify the Y axis in Fig. 1A and B.

Response: In this article, we have tested several antibodies using different methods, so the description of antibody values is placed in the materials and methods part.

-        Detail as the antibody titer was calculated. This data is missing in the cited ref indicated in the text.

Response: The Reed-Muench method was used to calculate the virus neutralization titers, which had been provided in the materials and methods part.

-        In fig. legend 1 specify if “antibody” refers to IgG.

Response: The “antibody” in the legend 1 is not only refers to IgG,also refers to neutralizing antibody, total antibody, and IgM by chemiluminescence immunoassay assays; nucleoprotein (N) antibody, and receptor-binding domain (RBD) antibody of SARS-CoV-2 by ELISA.

-        Specify which variant of omicron was tested.

Response: The variant of omicron tested is omicron BA.1, we had updated in the manuscript.

-        In fig. 2C-D some values are not readable for the presence of some dot.

Response: The figures in the article had been renewed.

-        Specify how flow cytometry gates were defined.

Response: We had defined the flow cytometry gates.

-        Specify concentration of BFA and SEB in materials and methods.

Response: We had added the concentration of BFA and SEB in materials and methods.

-        At line 265 use “SARS-CoV-2 Abs” instead of COVID-19 (that refers to the disease).

Response: We had changed the “COVID-19 Abs” to “SARS-CoV-2 Abs”.

-        At line 329 I suppose that “RPMI 1640” was the correct media. If yes, please correct.

Response: Yes, it is RPMI 1640.

-        Detail how spot of ELISPOT plated were read.

Response: The spots on the plates were analyzed by ELISPOT plate reader (AID Gmbh, Strassberg, Germany). We also had updated in materials and methods.

Reviewer 3 Report

General comment

The study "Immunity induced by inactivated SARS-CoV-2 vaccine: 2 Breadth, durability, potency, and specificity in a healthcare 3 worker cohort" is a potential interesting paper that address an important issue: how long the memory cells last and how these cells contribute to long-term immunological memory and protective immunity. The article is well written and presented but some issue need editorial review.

Major comments

The authors should give more details about how, where and when the sample was obtained and why the sample size was taken. They should explain how the individuals were asked to participate in the study and how they were followed.

Explain if previous COVID-19 infections were registered in each participants and if some of them became infected by the SARS-CoV-2 during the follow up

Minor comments

1) Ion Abstract: clearly specify the objective of the study

2) In Methods give more details about how the workers sample was taken.

3) Explain how previous infections in participants can affect the results..

4) Explain and discuss the main contribution of the study in the first paragraph of the Discussion

5) Comment on the implications of cellular immunity for deciding the type of vaccine to use and its effects on controlling the pandemic.

none

Author Response

Responses to Review 

General comment

The study "Immunity induced by inactivated SARS-CoV-2 vaccine: 2 Breadth, durability, potency, and specificity in a healthcare 3 worker cohort" is a potential interesting paper that address an important issue: how long the memory cells last and how these cells contribute to long-term immunological memory and protective immunity. The article is well written and presented but some issue need editorial review.

Major comments

The authors should give more details about how, where and when the sample was obtained and why the sample size was taken. They should explain how the individuals were asked to participate in the study and how they were followed.

Response: Thank you for your comments on the manuscript. All the participants were health-care workers from Zhejiang Hospital who had never been infected with SARSR-CoV-2 during the entire follow-up during the period from December 2020 to April 2022. All participants had been immunized with three doses of vaccines; the interval was 3 weeks between the first and second doses for nearly all participants. The third dose of vaccination was received an average of 274 days (range 146-291 days) after the two-dose vaccination. We drew blood from participants at these times: 2 weeks after first BBIBP-CorV immunization; 2 weeks and 6 months after the second immunization, and; 2 and 4 weeks, and 4 and 7 months after the third immunization (homologous booster).

Explain if previous COVID-19 infections were registered in each participants and if some of them became infected by the SARS-CoV-2 during the follow up

 Response: First of all, the participants in our study were all health-care workers in Zhejiang Hospital; Secondly, until termination of the study was terminated, the participant had tested the SARS-CoV-2 nucleic acid every other day. Therefore, all the subjects in this study were not infected with the SARS-CoV-2 before.

Minor comments

1) Ion Abstract: clearly specify the objective of the study

2) In Methods give more details about how the workers sample was taken.

3) Explain how previous infections in participants can affect the results..

4) Explain and discuss the main contribution of the study in the first paragraph of the Discussion

5) Comment on the implications of cellular immunity for deciding the type of vaccine to use and its effects on controlling the pandemic.

 Response: We had updated in the manuscript based on your comments.

Reviewer 4 Report

This manuscript gives an in-depth analysis of the types of immune response induced by an inactivated SARS-CoV2 vaccine over a period of 7 months.

The assays are well performed, and data clearly presented.

The main findings are that for sufficient protective response a booster vaccination is required for this particular vaccine. Also, that both B and T cell responses were initiated by the booster.

These findings will be of significant interest and warrant publication.

I do not feel that further experimental work or analysis is required, and the manuscript is clearly written.

However, before publication the figures need to be reproduced at a larger scale. In almost all, especially figures 2 – 4, the labelling is much too small to be legible on a standard printed page. This must be addressed before final publication.

Minor editorial checking required.

Author Response

Responses to Review 

This manuscript gives an in-depth analysis of the types of immune response induced by an inactivated SARS-CoV2 vaccine over a period of 7 months.

The assays are well performed, and data clearly presented.

The main findings are that for sufficient protective response a booster vaccination is required for this particular vaccine. Also, that both B and T cell responses were initiated by the booster.

These findings will be of significant interest and warrant publication.

I do not feel that further experimental work or analysis is required, and the manuscript is clearly written.

However, before publication the figures need to be reproduced at a larger scale. In almost all, especially figures 2 – 4, the labelling is much too small to be legible on a standard printed page. This must be addressed before final publication.

Response: Thanks for your interest and comments about this article. The figures2-4, we had been renewed in the article.